

# Artificial neural network with Taguchi method for robust classification model to improve classification accuracy of breast cancer

Md Akizur Rahman[1], Ravie chandren Muniyandi[1], Dheeb Albashish[2], Md Mokhlesur Rahman[1] and Opeyemi Lateef Usman[1]

[1] Research Center for Cyber Security, Faculty of Information Science and Technology, Universiti Kebangsaan Malaysia, 43600 Bangi, Selangor, Malaysia
[2] Computer Science Department, Prince Abdullah Bin Ghazi Faculty of Information and Communication Technology, Al-Balqa Applied University, Salt, Jordan

## ABSTRACT

Artificial neural networks (ANN) perform well in real-world classification problems. In this paper, a robust classification model using ANN was constructed to enhance the accuracy of breast cancer classification. The Taguchi method was used to determine the suitable number of neurons in a single hidden layer of the ANN. The selection of a suitable number of neurons helps to solve the overfitting problem by affecting the classification performance of an ANN. With this, a robust classification model was then built for breast cancer classification. Based on the Taguchi method results, the suitable number of neurons selected for the hidden layer in this study is 15, which was used for the training of the proposed ANN model. The developed model was benchmarked upon the Wisconsin Diagnostic Breast Cancer Dataset, popularly known as the UCI dataset. Finally, the proposed model was compared with seven other existing classification models, and it was confirmed that the model in this study had the best accuracy at breast cancer classification, at 98.8%. This confirmed that the proposed model significantly improved performance.

# INTRODUCTION

Despite current technological advances, medical sciences remain limited in their ability to contain and treat cancer diseases. The containment and treatment of cancer diseases form the crux of the medical science community's efforts at technological advancements. Cancer is known to be the most severe complex of diseases when it comes to mortality rates, and breast cancer is the most common leading cause of cancer death in women. Many women above 40 years old suffer from breast cancer. It is prevalent to the point that it has been identified as the second most deadly unavoidable disease for women in this age bracket (>40 years old) (*Imaginis, 2019*).

Corresponding author
Md Akizur Rahman,
akiz.akizur@gmail.com

In order to properly treat breast cancer, its identification and diagnosis during the early stages are crucial. The traditional approach to diagnosis is highly reliant upon the experience of the attending physician(s). The reliability of physicians' experience and visual inspections is questionable due to the high probability of human error. There is also an extremely large volume of datasets (big data) with poor quality and redundant information, making the diagnosis of cancer at an acceptable level of accuracy a complex affair despite physicians' vast experience. In an attempt to improve the accuracy of cancer classification, a computer-aided diagnostic (CAD) system has been used to assist physicians (*Sahran et al., 2018*; *Albashish et al., 2016*; *Xi et al., 2016*; *Aalaei et al., 2016*; *Rahman & Muniyandi, 2018b*; *Tomar & Agarwal, 2015*; *Elkhani & Muniyandi, 2017*; *Dheeba, Albert Singh & Tamil Selvi, 2014*; *Jafari-Marandi et al., 2018*). The use of the CAD system for classification is improving the medical diagnosis process. Classification systems can help not only to minimize possible mistakes associated with a lack of experience among physicians, but also to provide accurate information for the examination(s) of medical datasets (*Sahran et al., 2018*; *Rahman & Muniyandi, 2018a*).

Artificial neural network (ANN) models, which are inspired by the complex, interconnected neural structure of the brain, have been proposed for classification tasks. In ANN, learning is realized via experience, and behavior modifications are in response to environmental stimuli. The model also generalizes from previous examples to address new problems. Figure 1 shows the three layers of an ANN: input, hidden, and output. Two layers are in communication with the external environment. The input layer receives input signals directly from the environment, which are then processed by the network, while the output layer of the network delivers the processed results to the (outward) environment. The number of neurons in the output layer is directly linked to a particular number of tasks the neural network was designed to carry out. The intermediate layer linking the input and output layers is called the hidden layer. This layer contains a function known as the activation or transfer function, which performs nonlinear activation on the sum of the weighted inputs from the preceding layer. It is associated with the hidden neurons, which are neurons that are absent from both the input and output layers (*Karsoliya, 2012*). Hidden neurons result in two outcomes: overfitting and underfitting. The large number of hidden neurons can be a potential cause of the latter, while overfitting occurs when there are multiple unnecessary neurons present in the hidden layer (*Karsoliya, 2012*). Underfitting occurs when the number of hidden neurons is lower than what is required to model a problem dataset. The small number of neurons in the hidden layers are pressured to properly detect highly complex signals (*Karsoliya, 2012*). A neural network architecture is dictated by the number of its hidden layers, due to the fact that it is directly linked to the external environment (*Panchal, Ganatra et al., 2011*). The determination of the accurate number of neurons in hidden layer is crucial for increasing the accuracy of cancer classification. One method that is effective for the determination of the suitable number of neurons in hidden layer is the Taguchi method (*Wu & Wu, 2000*).

The Taguchi method (*Wu & Wu, 2000*) has gained prominence in several research works due to its focus on optimization problems. It performs well despite uncertain conditions, producing low-cost outputs and robust parameter design via the integration of

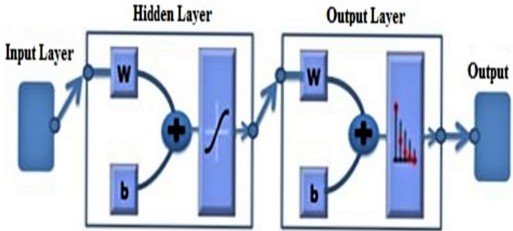

**Figure 1** The basic architecture of an ANN, which is consisted of the input layer, hidden layer and output layer.

traditional engineering with statistics for approximation and performance enhancement in multiple cases. For processes and parameter arrangement, the Taguchi method utilizes the statistical investigational architecture. In a robust experimental setup and design, processes or products can be investigated by altering design-related factors. This experimental design helps to proficiently and consistently analyze outputs (_Wu & Wu, 2000_; _Jaddi, Abdullah & Hamdan, 2013_).

Therefore, this study proposes the Taguchi method for the parameter optimization for an ANN algorithm, specifically for the determination of the optimal number of neurons in a single hidden layer. This helps to increase the accuracy of cancer classification.

The remainder of this paper is organized as follows: 'Related studies' introduces related works, while 'Materials & methods' details the experimental dataset, the Taguchi method, and ANN. 'The proposed artificial neural network model with 15 neurons in a single hidden layer for breast cancer classification' discusses the proposed algorithm for breast cancer classification. 'Experimental results and discussion' presents the experimental results and discussion. Finally, 'Conclusions' concludes the work and suggests future research possibilities.

## RELATED STUDIES

The most important requirements for machine learning techniques in medical diagnosis and cancer classification are accuracy and reliability. This section presents related studies that were previously conducted in this field, emphasizing ANN performance improvement and cancerous dataset classifications. The literature offers many examples of research works employing an experimental design to determine the suitable number of parameters, which could influence the performance of the ANN. To efficiently establish an ANN's parameters, _Khaw, Lim & Lim (1995)_ proposed the Taguchi method using two sets of simulated datasets in order to increase the accuracy and convergence speed of the back-propagation network (BPN). _Peterson et al. (1995)_ utilized the Taguchi method to determine the causes of faults in the BPN, while _Yang & Lee (1999)_ minimized the ANN training duration using the Taguchi method. _Packianather, Drake & Rowlands (2000)_ reported the outcome of parametric design on the performance of a neural network for a wood veneer examination using the Taguchi method. For the purpose of designing a multilayer feed-forward neural network, _Kim & Yum (2004)_ used Taguchi's active method to account for noise. _Tortum et_

al. (2007) utilized the Taguchi method to determine the suitable combination of effectual parameters, and reported the consequences of the performance criteria of every parameter in the neural network. Sukthomya and Tannock (*Bashiri & Farshbaf Geranmayeh, 2011*) utilized the Taguchi method to determine the suitable combination of effectual parameters in a neural network. *Jung & Yum (2011)* employed the Taguchi method to develop a dynamic parameter design that relies on an ANN. *Becherer et al. (2019)* used the parametric fine-tuning technique with a convolutional neural network (CNN) for the purpose of image classification.

On the topic of breast cancer diagnosis and classification, *Zheng, Yoon & Lam (2014)* proposed a hybridization of the k-means algorithm and support vector machine (K-SVM) for breast cancer diagnosis. The results achieved 97.38% accuracy using the Wisconsin Diagnostic Breast Cancer (WDBC) dataset. *Örkcü & Bal (2011)* compared the performance of a real-coded genetic algorithm, back-propagation neural network (BPNN), and binary coded genetic algorithm models using the breast cancer datasets, and reported accuracies of 96.50%, 93.10%, and 94.00%, respectively. *Salama & Abdelhalim (2012)* used the classifiers Naive Bayes (NB), sequential minimal optimization (SMO), decision tree (J48), multi-layer perception (MLP), and instance-based for k-nearest neighbor (IBK-NN) for breast cancer classifications. The experiment, which adopted a confusion matrix based on the 10-fold cross-validation method, using datasets from three distinct databases, showed that the highest classification accuracy of 97.70% was realized using the sequential minimal optimization (SMO) model.

On the other hand, *Malmir, Farokhi & Sabbaghi-Nadooshan (2013)* used an imperialist competitive algorithm (ICA) with multilayer perceptron (MLP) network and particle swarm optimization (PSO) for breast cancer classification. This study achieved 97.75% and 97.63% classification accuracies for MLP and PSO, respectively. *Koyuncu & Ceylan (2013)* attempted to achieve higher classification accuracy using a breast cancer dataset and Rotation Forest artificial neural network (RF-ANN) classifier. The result from the analysis of the above classifier was 98.05% accurate. *Aalaei et al. (2016)* compared the performance of a genetic algorithm with two similar classifiers, namely a particle swarm classifier and ANN, using the WDBC datasets without feature selection. The experiment reported accuracies of 96.40%, 96.50%, and 96.10%, respectively.

*Chaurasia & Pal (2017)* proposed Ensemble Boosting Learning (EBL) method for breast cancer classification. The author used UCI different breast cancer dataset including Wisconsin Breast Cancer Diagnostic (WBCD) dataset. The proposed method able to achieve the classification accuracy of 97.0% using 10-fold cross-validation with Radial Based Function Neural Network (RBFNN) classifier. *Xue, Zhang & Browne (2014)* proposed the PSO technique with novel initialization and updated mechanisms hybridized with a KNN classifier for breast cancer classification. Please replace the highlighted sentence. The authors implemented the proposed technique on the WDBC dataset partitioned into 70% for training and 30% for testing based on 10-fold cross-validation. The technique was able to achieve the classification accuracy off 92.98%. *Nekkaa & Boughaci (2015)* proposed a classification model by using the memetic algorithm (MA) with support vector machine (SVM) to address the classification problem. The authors used particular datasets to

compare certain popular classifiers including WDBC dataset. The model achieved 97.85% of classification accuracy.

*Ali, Hosni & Abnane (2020)* developed heterogeneous ensembles-based classification technique for breast cancer classification. The heterogeneous ensembles included support vector machines (SVM), multilayer perceptron (MLP), and decision trees (DTs) to evaluate the classification performance. The authors built three groups of heterogeneous ensembles using three single classifiers optimized by GS, PSO, and UC Weka. The proposed methods were implemented using UCI breast cancer dataset including WDBC and achieved the highest classification accuracy of 98.07% by GSVM. *Jijitha & Amudha (2020)* performed on six types of different breast cancer dataset including BCWD (Breast Cancer Wisconsin Diagnostic) dataset using machine learning techniques. For breast cancer classification the author used K-Nearest Neighbor (K-NN) and Logistic Regression (LR) technique. From BCWD dataset achieved a classification accuracy of 96.5% and 97.02% by LR and K-NN. *Thiyagarajan, Chakravarthy & Arivoli (2020)* investigated the application of machine learning methods for classification of breast cancer. For an investigation of breast cancer classification performance was used different machine learning methods which are included as follows: ANN, Decision Tree, KNN and SVM. The experiment was completed by using WBCD dataset and the highest classification accuracy of 96.2% was achieved from the ANN method. *Quy et al. (2020)* applied Machine Learning-Based Evolutionary Neural Network Approach for breast cancer classification. The author focused on parameter optimize of neural network (NN) model using Adaptive Particle Swarm Optimization (APSO) algorithm. The NN used 20 neurons in a single hidden layer. The WDBC dataset was used and partitioned into 70% for training and 30% for testing. The model was trained by Back-propagation (BP), classical PSO and APSO respectively. From the experimental result achieved the highest classification accuracy of 98.24% by APSO-NN.

From the literature, it can be concluded that ANN is utilized since it involves pattern recognition and data classification. The most important advantage of ANN with regard to the classification problem in multisource databases has been solved. ANN is an established tool in data classification that is easy to utilize and implement. In previous research, two limitations were identified: (1) low classification accuracy (*Salama & Abdelhalim, 2012*; *Chaurasia & Pal, 2017*; *Xue, Zhang & Browne, 2014*; *Nekkaa & Boughaci, 2015*), and (2) neuron selection in the hidden layer (*Aalaei et al., 2016*; *Quy et al., 2020*). This study proposes 15 neurons in single hidden layer of ANN which can assist to improve the accuracy of breast cancer classification.

## MATERIALS & METHODS

### Experimental dataset

This research used the Wisconsin Diagnostic Breast Cancer (WDBC) dataset from the UCI Machine Learning Repository (http://archive.ics.uci.edu/ml/datasets/Breast+Cancer+Wisconsin+(Diagnostic)) to differentiate malignant tumors from normal tumor samples. The dataset was used to compare normal tumors with cancerous (malignant) tumors (*Pobiruchin et al., 2016*). The dataset contains 32 features, namely ID, diagnosis,

and 30 real-valued input features, followed by 569 samples, of which 357 are normal and 212 are cancerous, with zero missing attribute values. The dataset features were originally computed from a digitized image of a fine needle aspirate (FNA) of a breast mass by the first user and have become a reference dataset for many recent studies on breast cancer classification. The attribute information is as follows:

## Attribute Information

| | |
|---|---|
| *1* | *ID number* |
| *2* | *Diagnosis (M = Malignant, N = Normal)* |
| 3 to 32 | Ten real-valued features are computed for each cell nucleus |
| *i* | *Radius (mean of distances from center to points on the perimeter)* |
| *ii* | *Texture (Standard deviation of gray-scale values)* |
| *iii* | *Perimeter* |
| *iv* | *Area* |
| *v* | *Smoothness (Local variation in radius lengths)* |
| *vi* | *Compactness (Perimeter$^2$ / area - 1.0)* |
| *vii* | *Concavity (Severity of concave portions of the contour)* |
| *viii* | *Concave points (Number of concave portions of the contour)* |
| *ix* | *Symmetry* |
| *x* | *Fractal dimension ("Coastline approximation" - 1)* |

For this study, our experimental dataset was divided into training, validation, and testing datasets using four different partitions based on the Taguchi method (Followed by Eq. (1) and Table 1). The partitions were chosen based on the experimental performance. The dataset partitions are as follows: The first partition contains 50, 25, and 25; the second partition contains 60, 20, and 20; the third partition contains 70, 15, and 15; and, finally, the fourth partition contains 80, 10, and 10, where the first number represents the training set, the second number represents the validation set, and the third number represents the testing set for each partition, respectively. The subsequent sections 3.2 and 4 present details of how the Taguchi method was utilized with it's performance on different dataset partitions. (Table 2).

## Taguchi Method

The Taguchi method is a robust experimental design (*Wu & Wu, 2000*) process that can be analyzed and improved upon by altering the relative design factors. It is also called the statistical method and can be used to realize the highest product quality. The Taguchi method utilizes a three-stage method: system design, parameter design, and tolerance design. In the system design, suitable working levels of design factors are accounted for. For design and testing, a system needs to be based on the designers' judgement of factors such as materials, parts, nominal products, processes, or parameters, based on the latest technology. The parameter design is used to determine parametric levels in order to enhance the accuracy of the process being considered. Tolerance design is used to fine-tune

**Table 1  Orthogonal array (OA).**

| RUN | F1 | F2 | F3 | F4 | F5 | F6 | F7 | F8 | F9 | F10 | F11 | F12 | F13 | F14 | F15 | F16 | F17 | F18 | F19 | F20 |
|---|---|---|---|---|---|---|---|---|---|---|---|---|---|---|---|---|---|---|---|---|
| 1 | 1 | 0 | 0 | 0 | 0 | 0 | 0 | 0 | 0 | 0 | 0 | 0 | 0 | 0 | 0 | 0 | 0 | 0 | 0 | 0 |
| 2 | 0 | 1 | 0 | 0 | 0 | 0 | 0 | 0 | 0 | 0 | 0 | 0 | 0 | 0 | 0 | 0 | 0 | 0 | 0 | 0 |
| 3 | 0 | 0 | 1 | 0 | 0 | 0 | 0 | 0 | 0 | 0 | 0 | 0 | 0 | 0 | 0 | 0 | 0 | 0 | 0 | 0 |
| 4 | 0 | 0 | 0 | 1 | 0 | 0 | 0 | 0 | 0 | 0 | 0 | 0 | 0 | 0 | 0 | 0 | 0 | 0 | 0 | 0 |
| 5 | 0 | 0 | 0 | 0 | 1 | 0 | 0 | 0 | 0 | 0 | 0 | 0 | 0 | 0 | 0 | 0 | 0 | 0 | 0 | 0 |
| 6 | 0 | 0 | 0 | 0 | 0 | 1 | 0 | 0 | 0 | 0 | 0 | 0 | 0 | 0 | 0 | 0 | 0 | 0 | 0 | 0 |
| 7 | 0 | 0 | 0 | 0 | 0 | 0 | 1 | 0 | 0 | 0 | 0 | 0 | 0 | 0 | 0 | 0 | 0 | 0 | 0 | 0 |
| 8 | 0 | 0 | 0 | 0 | 0 | 0 | 0 | 1 | 0 | 0 | 0 | 0 | 0 | 0 | 0 | 0 | 0 | 0 | 0 | 0 |
| 9 | 0 | 0 | 0 | 0 | 0 | 0 | 0 | 0 | 1 | 0 | 0 | 0 | 0 | 0 | 0 | 0 | 0 | 0 | 0 | 0 |
| 10 | 0 | 0 | 0 | 0 | 0 | 0 | 0 | 0 | 0 | 1 | 0 | 0 | 0 | 0 | 0 | 0 | 0 | 0 | 0 | 0 |
| 11 | 0 | 0 | 0 | 0 | 0 | 0 | 0 | 0 | 0 | 0 | 1 | 0 | 0 | 0 | 0 | 0 | 0 | 0 | 0 | 0 |
| 12 | 0 | 0 | 0 | 0 | 0 | 0 | 0 | 0 | 0 | 0 | 0 | 1 | 0 | 0 | 0 | 0 | 0 | 0 | 0 | 0 |
| 13 | 0 | 0 | 0 | 0 | 0 | 0 | 0 | 0 | 0 | 0 | 0 | 0 | 1 | | 0 | 0 | 0 | 0 | 0 | 0 |
| 14 | 0 | 0 | 0 | 0 | 0 | 0 | 0 | 0 | 0 | 0 | 0 | 0 | 0 | 1 | 0 | 0 | 0 | 0 | 0 | 0 |
| 15 | 0 | 0 | 0 | 0 | 0 | 0 | 0 | 0 | 0 | 0 | 0 | 0 | 0 | 0 | 1 | 0 | 0 | 0 | 0 | 0 |
| 16 | 0 | 0 | 0 | 0 | 0 | 0 | 0 | 0 | 0 | 0 | 0 | 0 | 0 | 0 | 0 | 1 | 0 | 0 | 0 | 0 |
| 17 | 0 | 0 | 0 | 0 | 0 | 0 | 0 | 0 | 0 | 0 | 0 | 0 | 0 | 0 | 0 | 0 | 1 | 0 | 0 | 0 |
| 18 | 0 | 0 | 0 | 0 | 0 | 0 | 0 | 0 | 0 | 0 | 0 | 0 | 0 | 0 | 0 | 0 | 0 | 1 | 0 | 0 |
| 19 | 0 | 0 | 0 | 0 | 0 | 0 | 0 | 0 | 0 | 0 | 0 | 0 | 0 | 0 | 0 | 0 | 0 | 0 | 1 | 0 |
| 20 | 0 | 0 | 0 | 0 | 0 | 0 | 0 | 0 | 0 | 0 | 0 | 0 | 0 | 0 | 0 | 0 | 0 | 0 | 0 | 1 |
| 21 | 0 | 0 | 0 | 0 | 0 | 0 | 0 | 0 | 0 | 0 | 0 | 0 | 0 | 0 | 0 | 0 | 0 | 0 | 1 | 0 |

the results. As a commonly used robust design approach, the Taguchi method has two mechanisms: Orthogonal array and signal-to-noise ratio (SNR) (*Chuang et al., 2010*), for improvement and analysis. To minimize experimental efforts, Orthogonal array is mainly utilized, using $N$ number of design parameters.

An Orthogonal array is very useful, as it gives an extensive investigation of associations among all design factors and reasonably adjusts the methodical correlations of various dimensions of each factor. An orthogonal array is a two-dimensional array. Each column represents a certain design parameter, while each row denotes an experimental test with an actual arrangement of various levels for all of the design factors. In this research, the two-level Orthogonal array for determining optimal neurons is shown in Eq. (1):

$$L_M(2^N), \tag{1}$$

where $N$ is the number of columns in the Orthogonal matrix. $M = 2K$ ($M > N$, $K > \log_2(N)$; $M$ is the number of expected experimental trials, and $K$ is an integer. Base 2 is the number of levels of every design parameter. In this study, there are 20 numbers of columns in the Orthogonal matrix (where F1 to F20 indicate the design factor the number of neurons in the hidden layer), and a two-level Orthogonal array was used for selecting appropriate number of neurons. The two-level Orthogonal array was created using $L_{21}(2^{20})$, as shown in Table 1.

**Table 2 The Taguchi method for ANN to select the suitable number of neurons for breast cancer datasets.**

| First | Training accuracy (%) | Validation accuracy (%) | Data partitioning | | |
|---|---|---|---|---|---|
| **Number of neurons selected by Taguchi method** | | | Training (%) | Validation (%) | Testing (%) |
| 10 | 89.7 | 89.4 | | | |
| 11 | 89.9 | 90.8 | | | |
| 12 | 90.8 | 92.2 | | | |
| 13 | 91.5 | 92.5 | | | |
| 14 | 92.7 | 92.8 | | | |
| **15** | **93.5** | **93.9** | 50 | 25 | 25 |
| 16 | 93.0 | 93.4 | | | |
| 17 | 92.8 | 91.9 | | | |
| 18 | 92.1 | 92.3 | | | |
| 19 | 91.2 | 91.7 | | | |
| 20 | 90.9 | 91.3 | | | |
| Second | Training Accuracy (%) | Validation Accuracy (%) | Data partition | | |
| Number of neurons selected by Taguchi method | | | Training (%) | Validation (%) | Testing (%) |
| 20 | 90.7 | 91.1 | | | |
| 19 | 90.1 | 90.8 | | | |
| 18 | 91.9 | 91.2 | | | |
| 17 | 93.7 | 92.8 | | | |
| 16 | 95.2 | 94.7 | | | |
| **15** | **96.6** | **96.2** | 60 | 20 | 20 |
| 14 | 96.1 | 96.2 | | | |
| 13 | 95.8 | 95.6 | | | |
| 12 | 95.4 | 95.3 | | | |
| 11 | 93.8 | 94.2 | | | |
| 10 | 92.3 | 93.0 | | | |
| Third | Training Accuracy (%) | Validation Accuracy (%) | Data partition | | |
| Number of neurons selected by Taguchi method | | | Training (%) | Validation (%) | Testing (%) |
| 10 | 93.6 | 93.1 | | | |
| 11 | 94.5 | 94.3 | | | |
| 12 | 95.7 | 94.9 | | | |
| 13 | 97.2 | 96.8 | | | |
| 14 | 98.0 | 97.5 | | | |
| **15** | **98.5** | **98.3** | **70** | **15** | **15** |
| 16 | 98.3 | 98.2 | | | |
| 17 | 97.8 | 97.6 | | | |
| 18 | 97.1 | 97.1 | | | |
| 19 | 96.8 | 96.3 | | | |
| 20 | 96.2 | 96.3 | | | |

**Table 2** (*continued*)

| Fourth | Training Accuracy (%) | Validation Accuracy (%) | Data partition | | |
|---|---|---|---|---|---|
| Number of neurons selected by Taguchi method | | | Training (%) | Validation (%) | Testing (%) |
| 20 | 93.4 | 93.1 | | | |
| 19 | 92.8 | 93.2 | | | |
| 18 | 92.2 | 92.9 | | | |
| 17 | 93.1 | 92.7 | | | |
| 16 | 93.9 | 92.4 | | | |
| **15** | **94.7** | **93.8** | 80 | 10 | 10 |
| 14 | 94.2 | 93.4 | | | |
| 13 | 93.9 | 93.8 | | | |
| 12 | 93.2 | 93.6 | | | |
| 11 | 93.0 | 92.5 | | | |
| 10 | 92.9 | 92.4 | | | |

Only 21 experimental trials are required for evaluation, analysis, and improvement. Conversely, all possible combinations of 20 design factors (i.e., $2^{20} = 1,048,576$) should be accounted for in the full factorial experimental design, which is frequently inapplicable in practice (*Yang et al., 2008*) (for neuron selection: 1: selected, 0: not selected).

This research used an orthogonal array mechanism to analyze and enhance the ANN algorithm performance by determining the optimal number of neurons in hidden layer. If a particular target has $N$ different design factors, $2^N$ possible experimental trials will be considered in the full factorial experimental design.

## Artificial neural network

An artificial neural network (ANN) (*Rahman & Muniyandi, 2018a*) is a machine learning approach, which models the human brain with a number of artificial neurons and interconnected associations. The neurons in ANNs tend to have fewer connections relative to a biological neural system. Figure 1 illustrates the basics of the ANN architecture.

Neurons are highly interconnected computational units inspired by the mammalian brain. The ANN system consists of the smallest processing nodes called neurons, or processing elements. To obtain outputs, a function called Sigmoid activation works on inputs and connects the weights with the help of neurons. The connections between weight (w) values and single nodes are called biases (b). The iterative flow of training data determines weight values throughout the network. During the training phase, weight values are verified until the network is able to detect a particular cluster using criteria for typical input data. Also, these weights assign the link relating one layer of neurons to another. Hence, changes in the relationship between input and output take place repeatedly, along with changes in weight values. The method of balancing the links' weight values by repeatedly exposing the network to the input–output dataset for learning is called training.

ANN has various types of architectures. Multi-layer feed-forward neural networks have been widely used in cancer classification (*Alesawy & Muniyandi, 2016*; *Hopfield, 1988*; *Bebis & Georgiopoulos, 1994*). The architecture of a Multi-layer feed-forward neural network system is shown in Fig. 2, consisting of one or more input layer and one or more hidden layer along with an output layer.

## THE PROPOSED ARTIFICIAL NEURAL NETWORK MODEL WITH 15 NEURONS IN A SINGLE HIDDEN LAYER FOR BREAST CANCER CLASSIFICATION

In this section, an ANN model with 15 neurons in a single hidden layer is presented for breast cancer classification. The ANN model, which applied the Taguchi method (*Wu & Wu, 2000*) to determine the optimal number of neurons in the hidden layer for breast cancer classification, is called the Improved-ANN (IANN). The design of the proposed IANN model consists of six phases: data collection, data preprocessing, data loading using MATLAB simulation tool, data partitioning, ANN training and validation, and the application of the Taguchi method to determine the optimal number of neurons in the hidden layer of the ANN.

The first step is data collection. It involves the collection of the appropriate dataset from the data source and center for the purpose of research. In the second step (data preprocessing), all of the data features were prepared and filtered to remove noise in order to enhance the quality of the features that will be selected for the classification exercise. In the third stage, the Neural Network Pattern Recognition Tool of MATLAB (R2017b) simulation software (*Mathworks, 2017*) was used to load the dataset. Then, the dataset was divided into three categories: a training dataset, a validation dataset, and a testing dataset. A two-layer feed-forward neural network, Gradient Descent withSigmoid activation function and Softmax Output Neurons (patternnet), was used to classify the dataset with adequate neurons in the hidden layer. Multi-layer feed-forward neural network is widely used for classification problem and it is able to achieved high classification performance. The most algorithm to train neural network is used gradient descent. Gradient descent is a way to minimize an objective function which assist to adapt the learning rate. The sigmoid activation function is a nonlinear function which can help the network learn complex data in hidden layer. The softmax function is able to handle multiple classes output. Basically, Softmax function is used for output layer. The number of neurons for the experiments was at least 10 (default value) because below that the performance was too low. It could be seen that, in the case of each training phase, the performance of the model increased gradually and peaked when the number of neurons in the hidden layer hit 15. Further increasing the number of hidden layer neurons from 16 to 20 neurons decline the observed performance. The number of neurons determined using the Taguchi method is detailed in 'Experimental setup and evolution methods' and Table 2. By this method, the number of hidden layer neurons was selected randomly and repeatedly until excellent accuracy was obtained for the validation set. The excellent performance of the validation set in this study was achieved via 15 neurons in the hidden layer, which was sufficient to construct

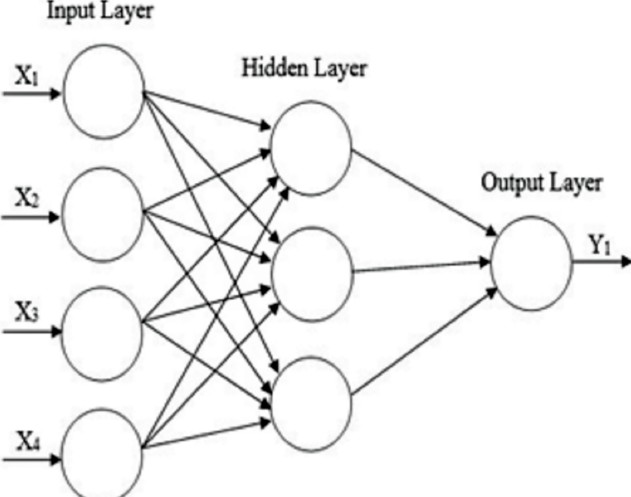

**Figure 2** The architecture of a Multi-layer feed-forward neural network with four inputs (features), one hidden layer, and one output layer.

a suitable ANN model for breast cancer classification when the number of output layer neurons was two, as shown in Fig. 3.

Understanding the classification of breast cancer forms the first step in its containment and eradication worldwide. Cancer researchers have utilized various machine learning algorithms for cancer classification. The most significant benefit of ANN for the classification of the problem in multisource databases has been solved (*Cai & Jiang, 2014*; *Wadhonkar, Tijare & Integration, 2014*; *Rahman & Muniyandi, 2020*; *Huang et al., 2014*). This study employs an ANN with 15 neurons in hidden layer for cancer classification. Figure 4 shows the conceptual framework for breast cancer classification using an ANN model with 15 neurons in its hidden layer.

# EXPERIMENTAL RESULTS AND DISCUSSION

## Experimental setup and evolution methods

In this research, the experiment was conducted by using MATLAB Tool, WDBC dataset, Taguchi Method and ANN Algorithm, implemented on a system with the following configuration: Core i7 GPU, Windows 10 with 8GB RAM and 1TB HDD. The proposed ANN model comprises a single hidden layer with 15 neurons carefully selected using Taguchi method. Accuracy of the proposed model was obtained directly from the confusion matrix using the formula:

$$\text{Accuracy} = \frac{TP+TN}{TP+TN+FP+FN} \times 100.$$

## ANN parameter optimization based on the Taguchi Method

To build an appropriate classification model using ANN, parameter selection is one of the most significant steps. The performance and stability of an ANN is dictated by its selected

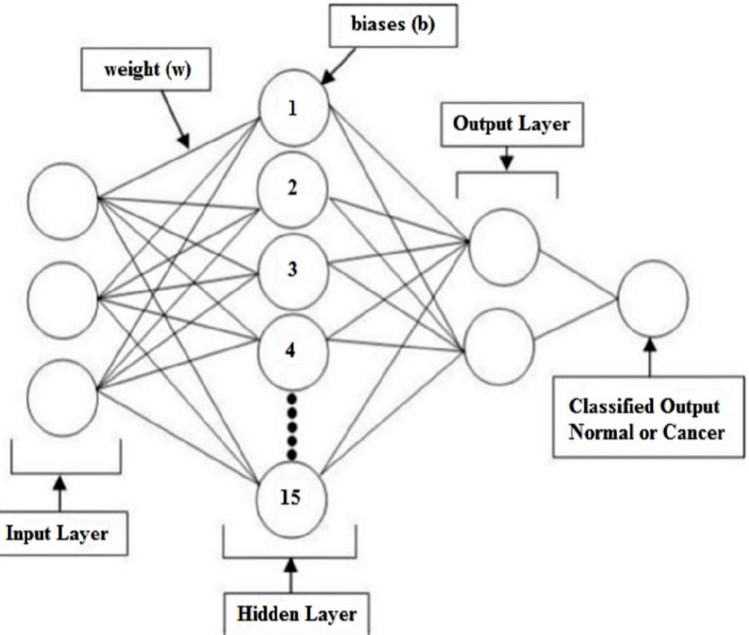

**Figure 3** **Architecture of the proposed 15 neurons ANN classifier with two output layers.** The proposed ANN classifier includes input layer, hidden layer and output layer. Where in hidden layer 15 neurons are utilized while the two output layer produce the final outcome result which is either normal or cancer.

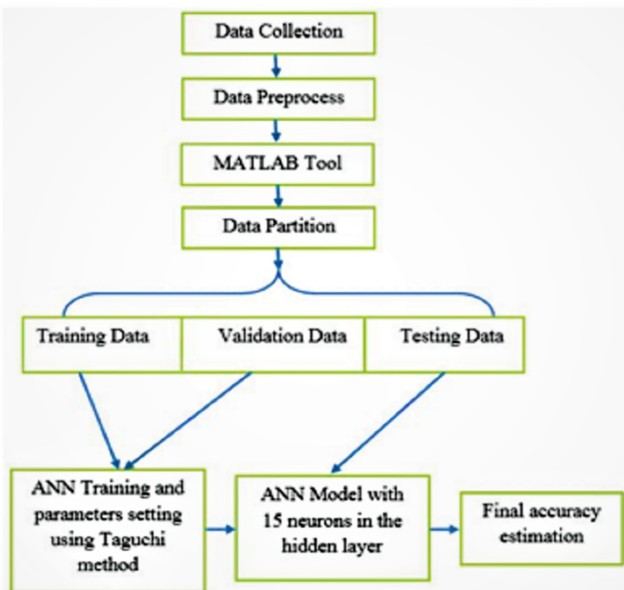

**Figure 4** **The proposed Improved-ANN (IANN) cancer classification model.** It starts by data collection, followed by the preprocess step to clean the data. Then, the data is divided into training, validation, and testing. Then, the ANN is trained and the parameter is optimized using Taguchi method. The final ANN model with 15 neurons in the hidden layer is used to classify the test (unseen) data.

parameters. The Taguchi method is mainly used for experimental parametric selection, owing to the concept of engineering and technology. For instance, this method can decrease the vast number of experiments and simultaneously analyze several parameters. For excellent performance and low-cost computations, this method performs well in the context of systematic and efficient designs (*Huang et al., 2014*). A different partition for a dataset can be created to simplify the selection of a suitable dataset.

As per 'Experimental dataset', the $L_{21}$ Orthogonal array was used to determine the appropriate number of neurons in the hidden layer of the proposed IANN model. The number of neurons for the experiments ranges from 10 (the default value) to 20, as shown in Table 2. Also, different partitions of the dataset were utilized in order to determine a suitable partition for the breast cancer dataset. For the different partitions of the dataset, Taguchi Orthogonal array mechanism was implemented in the same way, where $N = 3$ is the number of design factors of dataset partitions. Therefore, the $L_4$ Orthogonal array was used for data partition (for more details, see 'Experimental dataset', Eq. (1)). The dataset was randomly divided into three partitions: training, validation, and testing. These partitions were utilized in different groups: first, 50%, 25%, and 25%; second, 60%, 20%, and 20%; third, 70%, 15%, and 15%; and fourth, 80%, 10%, and 10%.

It can be observed in Table 2 that the hidden layer parameter is crucial to the performance of the ANN. As established earlier, the experiments were conducted in four phases using different dataset partitions for each phase. For the first experiment, we partitioned the entire dataset into 50% training, 25% validation, and the remaining 25% for testing. Based on the three partitions used, the highest accuracy for the validation dataset was realized when the number of neurons in a single hidden layer is 15. In this experiment, the accuracy was found to be 93.90% for the validation dataset and 93.50% for the training dataset. However, when the number of neurons increased, the performance diminished significantly. This seemingly implied that when the number of neurons in the hidden layer increases further, the performance of the ANN model is negatively affected.

In the second experiment, when the partitioning of the dataset was changed to 60%, 20%, and 20% for training, validation, and testing, respectively, the Taguchi method confirmed that the highest performance was maintained at 15 neurons in a single hidden layer. It is therefore clear from Table 2 that utilizing the aforementioned partitions significantly improves the performance of IANN model by ~3.0%.

In the third experiment, a new partition for the dataset was introduced, with 70% data for training, 15% for validation, and 15% for testing. In this scenario, the highest accuracy for the validation dataset was determined to be 98.30%, and that of the training dataset was 98.50% when the number of neurons in a single hidden layer was 15 using the Taguchi method.

Finally, in the fourth experiment, the partitioning of the dataset was 80%, 10%, and 10% for training, validation, and testing, respectively. This study confirmed that the results declined from 98.30% for the validation dataset to 93.80% with 4.5% when compared with the third experiment. All of the experiments proved that, 15 neurons are sufficient for modeling a single hidden layer in the proposed IANN for breast cancer classification.

From the empirical results, this study posits that the optimum conditions for breast cancer classification are 15 neurons in a single hidden layer and the partitioning of the data into 70%, 15%, and 15% for training, validation, and testing sets, respectively. Thus, we built an IANN model with three layers: input, hidden, and output. In summary, there were 31 input neurons at the input layer and 2 output neurons at the output layer, while the number of neurons in a single hidden layer was 15.

## Breast cancer classification results

As established in 'The proposed artificial neural network model with 15 neurons in a single hidden layer for breast cancer classification', the suitable number of neurons for a single hidden layer based on the Taguchi is 15. In addition, 70% of the dataset for training, 15% for validation, and 15% for testing is the optimal partitioning. The experiments were simulated on MATLAB (R2017b) software using Neural Network Toolbox for the implementation of the proposed model. The proposed IANN model has been tested using the WDBC dataset.

Table 3 shows the results from 30 different classification simulations using the WDBC dataset with their corresponding percentage errors. The average of the tests' accuracy was 98.80%. This study confirmed that the results of the simulations were relatively similar, which could be due to the high stability resulting from the selection of suitable parameters for the IANN classifier.

This study used 569-sample datasets. Each data sample consists of 32 features, with a total of 18,208 data points distributed as follows: 11,424 data points for normal and 6784 data points for cancerous. These features were used to train and simulate the proposed IANN model. A hidden-layer simple feed-forward neural network architecture is considered because our aim is to enhance the classification of breast cancer through optimal neuron selection in the hidden layer. The proposed IANN was tested with training, validation, and testing sets from the dataset. The network was adjusted based on the reported error during testing. Validation was used to simplify the network and halt training. Testing is ineffective in training; therefore, the performance of the network provides independent measures during and after training. A breast cancer dataset consisting of 569 samples was divided randomly into two groups: 399 samples (70%) for training and 170 samples (30%) for testing. The dataset for training was randomly divided into three groups: out of 399 samples 279 (70%) samples for training, and 60 (15%) samples each for testing and validation respectively.

Figures 5–11 show the IANN training performance, training state performance, error histogram, performance confusion matrix, ROC curve and testing dataset results for breast cancer, respectively.

It can be seen in Fig. 5 that the IANN training performance plot at the beginning of the training of cross-entropy resulted in the maximum error. The proposed system reported the best performance at epoch 20 iterations and an exact cross-entropy of 0.031613.

Figure 6 shows the network training state performance at epoch 26, when the gradient is 0.032875. The network halts the training session because its generalization stops improving.

**Table 3  Classification accuracy of the proposed method based on 30 different runs.**

| Number of experiments | Training accuracy (%) | Validation accuracy (%) | Testing accuracy (%) | Percent error |
|---|---|---|---|---|
| 1 | 97.7 | 96.5 | 98.7 | 1.3 |
| 2 | 98.5 | 98.7 | 98.8 | 1.2 |
| 3 | 98.9 | 97.9 | 99.2 | 0.8 |
| 4 | 98.2 | 98.7 | 98.5 | 1.5 |
| 5 | 97.7 | 97.2 | 98.6 | 1.4 |
| 6 | 98.6 | 98.6 | 98.8 | 1.2 |
| 7 | 98.8 | 98.5 | 99.2 | 0.8 |
| 8 | 98.4 | 98.2 | 98.4 | 1.6 |
| 9 | 98.7 | 98.7 | 99.2 | 0.8 |
| 10 | 98.5 | 98.5 | 98.8 | 1.2 |
| 11 | 98.2 | 98.2 | 98.8 | 1.2 |
| 12 | 98.8 | 98.8 | 99.2 | 0.8 |
| 13 | 98.3 | 98.0 | 98.6 | 1.4 |
| 14 | 98.5 | 98.3 | 98.8 | 1.2 |
| 15 | 98.3 | 98.5 | 98.7 | 1.3 |
| 16 | 98.5 | 98.5 | 98.9 | 1.1 |
| 17 | 98.6 | 98.5 | 99.2 | 0.8 |
| 18 | 98.8 | 98.5 | 99.2 | 0.8 |
| 19 | 98.9 | 98.7 | 98.8 | 1.2 |
| 20 | 98.8 | 98.5 | 98.9 | 1.1 |
| 21 | 98.2 | 98.2 | 98.5 | 1.5 |
| 22 | 98.5 | 98.5 | 98.7 | 1.3 |
| 23 | 98.9 | 98.8 | 98.8 | 1.2 |
| 24 | 98.6 | 98.5 | 98.3 | 1.7 |
| 25 | 98.5 | 98.5 | 98.8 | 1.2 |
| 26 | 98.4 | 98.4 | 99.0 | 1.0 |
| 27 | 98.9 | 98.5 | 99.1 | 0.9 |
| 28 | 98.6 | 98.1 | 98.4 | 1.6 |
| 29 | 98.2 | 98.3 | 98.9 | 1.1 |
| 30 | 98.6 | 97.5 | 98.6 | 1.4 |
| **Average** | **98.5** | **98.3** | **98.8** | **1.2** |

Error histograms for training, validation, and testing data are shown with 20 bins in Fig. 7. The experimental results shown in error histogram of Fig. 7 indicate that, the proposed system can handle the dataset used successfully, since the error is close to zero.

To evaluate the accuracy, a confusion matrix was used for all partitions of the dataset. A confusion matrix is a two-dimensional array, $r \times r$ (where $r$ is the number of classes). Figure 8 shows the MATLAB output representation of training dataset confusion matrix performance after 10-fold cross-validation. In Fig. 8, the first two rows and first two columns represent the actual confusion matrix. The third row and third column are the summary of percentage accuracy, and sensitivity and specificity, respectively. Row one, column one is the true positives (TP); row one, column two is the false negatives (FN); row

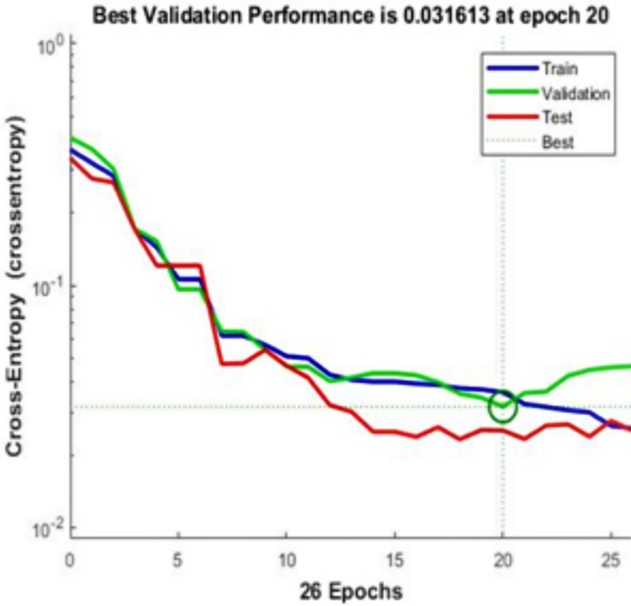

**Figure 5  IANN training performance.** The IANN is trained using 26 epochs (iterations) and using cross-entropy as performance measurement. This figure, shows the results of the training model. The best result is achieved at 20 epoch iteration with an exact cross-entropy of 0.031613

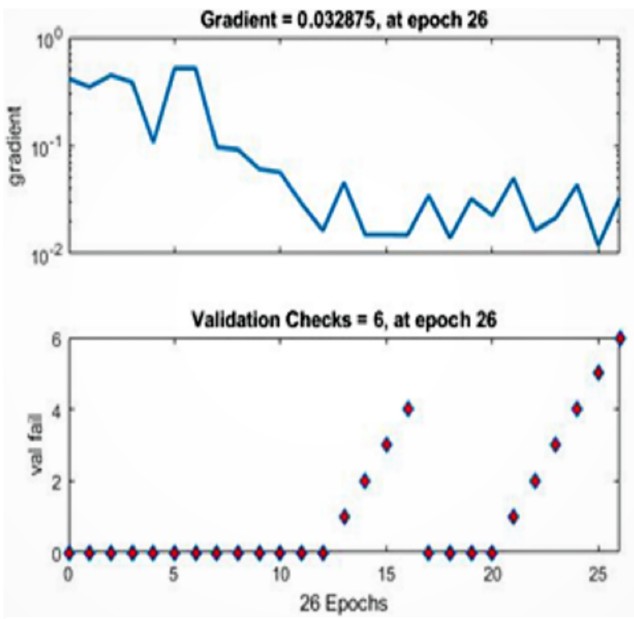

**Figure 6  Training state performance.** Based on this figure, the best validation performance is achieved at epoch 26 where the gradient is 0.032875.

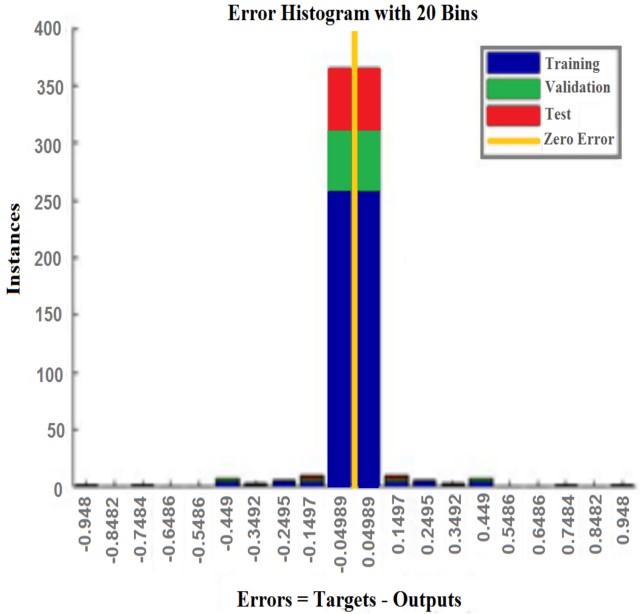

**Error Histogram with 20 Bins**

**Figure 7** **Error histogram.** This figure shows the visualize errors between the training, validation, and testing data using 20 bins. The visualize error show that the error is close to zero, which indicates the goodness of the proposed system.

two, column one is the false positives (FP); and row two, column two is the true negatives, respectively. All false positives (FP) are Type I errors, while all false negative (FN) are Type II errors generated by different classifiers.

The first two diagonal cells of the confusion matrix demonstrate the correct classification number and percentage accuracy of the trained network. The normal biopsy results are 247, which were correctly classified to represent 61.90% of the total 399 biopsies. On the other hand, the cancerous (malignant) biopsy results show 146 correctly classified tumors, representing 36.60% of the total biopsies. The experiment also revealed that three of the cancerous biopsies were incorrectly classified as normal, representing 0.8% of the total biopsy dataset, while three of the normal biopsies were incorrectly classified as cancerous, also representing 0.8% of the total biopsy data. The total result of 250 normal revealed that 98.80% was correct, and 1.20% incorrect. The total result of 149 cancerous was 98.00% correct and 2.00% incorrect. The total result of 250 normal cases was 98.80% correctly classified as normal, and 1.2% classified as cancerous. Out of 149 cancerous cases, 98.00% were correctly classified as cancer, and 2.00% classified as normal. The confusion matrix plots with 98.50% accuracy show that this system performed well and had 1.40% misclassification during its training stage from the proposed IANN.

The neural network training performance with receiver operating characteristic (ROC) plot is shown in Fig. 9. The ROC plot represents the performance of the binary classification system when the discrimination threshold fluctuates. The graph is formed by plotting the true positive rate (TPR) against the false positive rate (FPR). From ROC plot, it can be seen

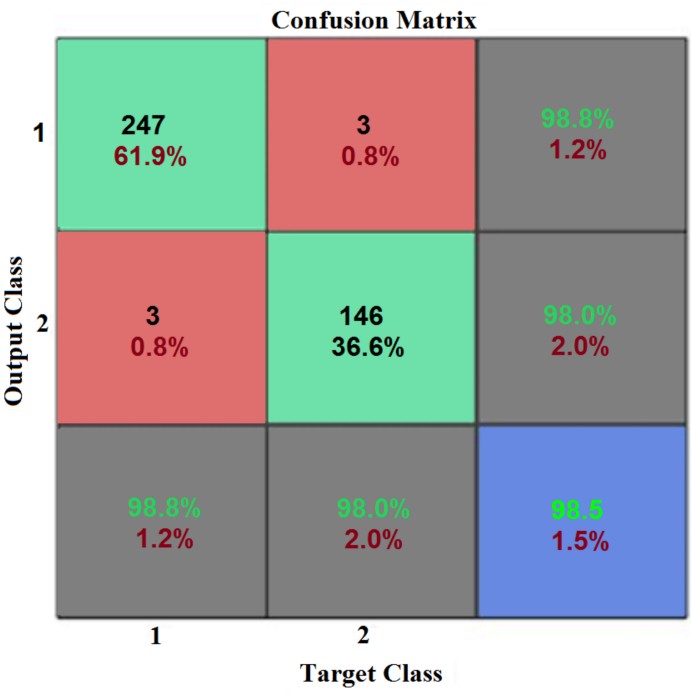

**Figure 8  Performance confusion matrix (CM).** This CM is computed for the performance of the training dataset. The first two rows and first two columns represent the actual confusion matrix. While, the third row and third column are the summary of percentage accuracy, and sensitivity and specificity, respectively.

that the NN's performance increases in response to the number of iterations. A perfect classification result was evident at 26 iterations. This shows that every class achieved perfect classification accuracy. Iteration 26 is the optimal iteration for the proposed model where the IANN performed at its peak.

After completing the training session, we tested the network for accurate classification, using 30% of the test dataset of breast cancer. Figure 10 shows the testing accuracy, otherwise known as the classification accuracy from the proposed IANN, to be rather high.

Figure 10 shows the experimental results with a test dataset of breast cancer: normal versus malignant tumor is correctly classified at 98.8% accuracy for the test case.

This study computed the receiver operating curve (ROC) and the results are shown in Fig. 11. The ROC curve indicates TPR and FPR at different edge settings of the network, with better results arising from the proposed system. The IANN, after training, validation, and testing, achieved 98.8% correct classification for two classes: normal and cancerous (malignant). The area under the curve was large.

Table 4 shows the model performance obtained from the confusion matrix shown in Figs. 8 and 10. In Table 4, the F-Score for the training dataset is 98.4%, while the F-Score for the testing dataset is 98.8%.

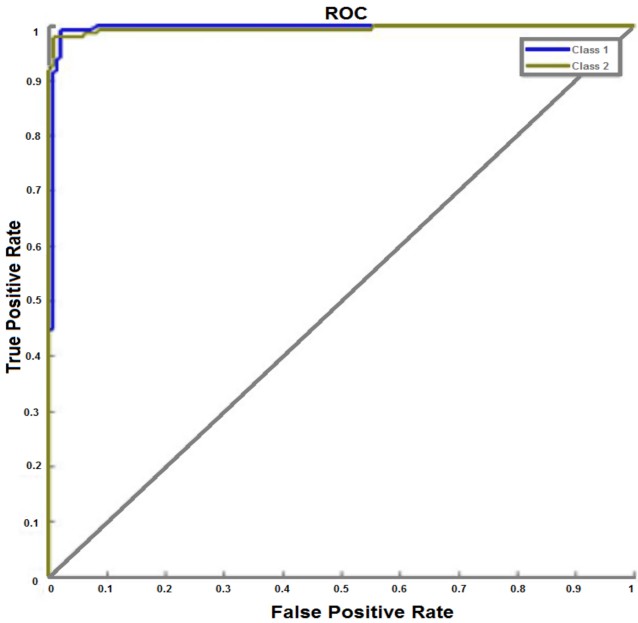

**Figure 9** **Receiver operating characteristic (ROC) curve.** The ROC plot represents the performance of the binary classification system for the training dataset. The graph is formed by plotting the true positive rate (TPR) against the false positive rate (FPR). This figure shows the best performance for the proposed IANN which is achieved at iteration 26.

## Discussion of findings and implications

Table 5 shows the performance of multiple machine learning algorithms relative to the proposed IANN model. Performance comparison was realized via MATLAB Classification learner tool-staking for the breast cancer dataset. Although the results obtained with other machine learning algorithms are equally good, the result reported by our model is optimal. It can be surmised that the proposed IANN outperforms the existing machine learning models in classifying breast cancer.

Table 6 indicates the superiority of this study in the context of cancer classification based on the same dataset but different methods for classification. Among the previous studies tabulated in Table 6, a lower classification accuracy of 92.98% was achieved by *Xue, Zhang & Browne (2014)*. These authors proposed the PSO technique with novel initialization and updated mechanisms hybridized with a KNN classifier and 10-fold cross-validation to maximize the classification performance of the WDBC datasets and dataset were partitioned for training 70% and 30% for testing. *Quy et al. (2020)* achieved the best classification accuracy of 98.24% using APSO-NN. For the NN used 20 neurons in a single hidden layer. The WDBC dataset was used and partitioned 70% for training and 30% for testing. *Abdar et al. (2020)* used the WDBC dataset for the experiment and achieved a classification accuracy of 98.07% using SV-NB-3-MetaClassifiers with a K-fold cross-validation technique. The performance is quite good. *Nekkaa & Boughaci (2015)* used a memetic algorithm (MA) with support vector machine (SVM) to address the classification problem. The authors used particular datasets to compare certain popular classifiers for

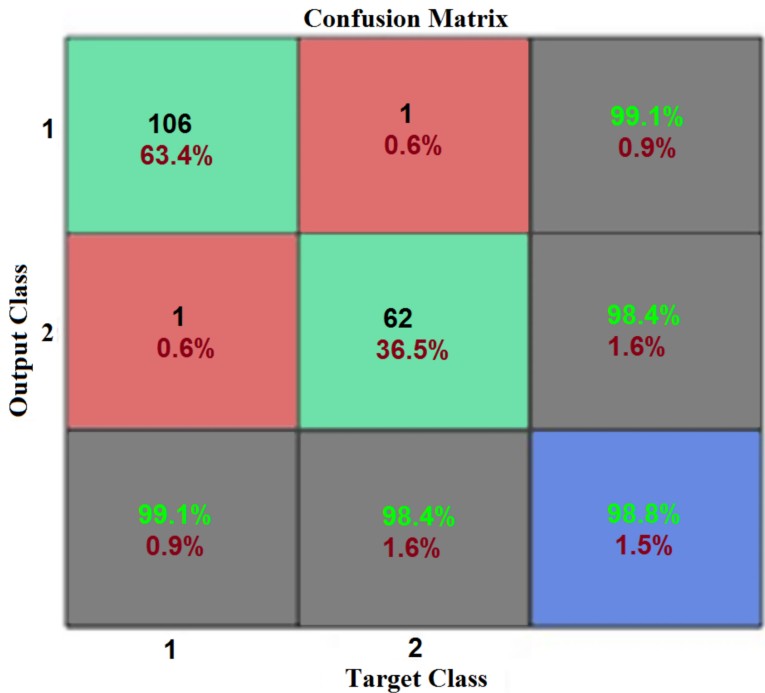

**Figure 10 Test dataset result of breast cancer.** This confusion matrix shows the performance of the proposed IANN on the test dataset (unseen). The first two rows and first two columns represent the actual confusion matrix. While, the third row and third column are the summary of percentage accuracy, and sensitivity and specificity, respectively.

**Table 4 Model performance based on confusion matrix after 10-fold cross-validation.**

| Dataset | Accuracy (%) | Sensitivity (%) | Specificity (%) | F-Score (%) |
|---|---|---|---|---|
| Training Dataset | 98.5 | 98.8 | 98.0 | 98.4 |
| Testing Dataset | 98.8 | 99.1 | 98.4 | 98.8 |

the data classification task. The model achieved 97.85% accuracy using WDBC datasets. Beside, *Salama & Abdelhalim (2012)* achieved a classification accuracy of 97.70% using sequential minimal optimization (SMO) technique with a confusion matrix based on the 10-fold cross-validation method. *Amrane et al. (2018)* used the KNN and Naive Bayes (NB) classifier machine learning technique for breast cancer classification. The author partitioned the dataset 60% for training and 40% for tesing. From KNN classifier achieved the highest classification accuracy of 97.51% which is not so high. *Aalaei et al. (2016)* used a GA-based classifier with ANN and reported a classification accuracy of 96.50% by making use of a single hidden layer with 5-neurons and 2-neurons in output layer, and the dataset partitioned of 80% for training and 20% for test data.

From the analysis of the results presented in Tables 5 and 6, it is obvious that the proposed method outperformed the other methods due to it realizing a classification accuracy of 98.80% —an improvement of 0.56% compared to *Quy et al. (2020)*, which has the best

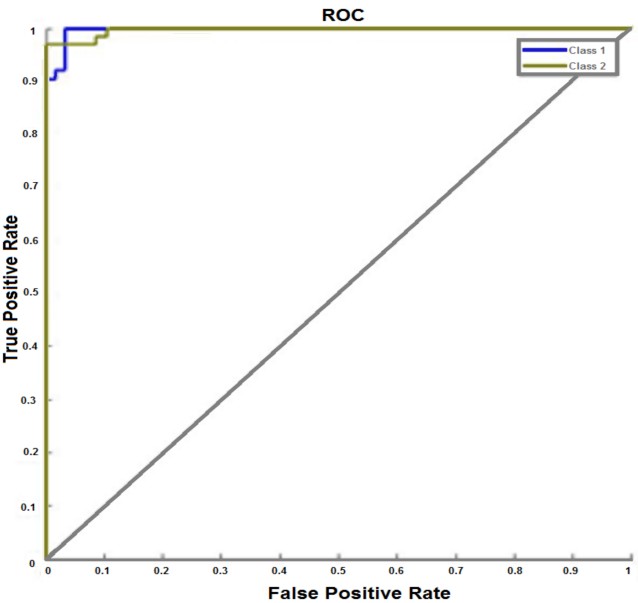

**Figure 11** **ROC curve with test dataset of breast cancer.** The ROC curve indicates TPR and FPR at different edge settings of the network. This figure shows how the proposed IANN achieved 98.8% correct classification for two classes: normal and cancerous (malignant) on test dataset.

performance among the existing methods. This could be due to the use of the Taguchi method for selecting the optimal number of neurons in a single hidden layer of ANN and suitable percentage data partition. Our results also show that the proposed method is more stable and reliable than existing classification models. By implication, incorporating IANN into breast cancer prediction could enable timely and accurate prediction of breast cancer, thereby helping medical practitioners to make the most appropriate decisions on breast cancer treatment. Table 7 (where $K = 10$) shows the 10-fold cross validation experimental performance which was conducted to validate the results shown in Table 5 and Table 6.

## CONCLUSIONS

In breast cancer diagnosis, differentiating between normal and malignant tumors is one of the challenges faced by physicians. In order to tackle this problem, ANN is incorporated in the CAD system for the binary classification of breast cancer datasets into normal/malignant tumors. The main purpose of this research is to enhance the classification accuracy to improve image-based diagnosis. The proposed IANN utilized the Taguchi method to optimize the hidden layer parameter of the ANN model. The experimental results proved that 15 is the optimal number of neurons for a single hidden layer of ANN, and can enhance the classification accuracy for the training, testing, and validation of selected datasets.

In this paper, the proposed IANN method reported 98.80% classification accuracy for breast cancer dataset classification. Additionally, 10-folds cross validation experiments performed (shown in Table 7) also confirmed the efficiency of the proposed IANN model with an average accuracy of 98.7% on testing dataset. The empirical results, presented

**Table 5** Performance comparison between different classifiers and the proposed method in breast cancer classification.

| Model | Name of the classifier | Accuracy |
|---|---|---|
| 1.1 | Fine Tree | 97.20% |
| 1.2 | Medium Tree | 96.90% |
| 1.3 | Coarse Tree | 97.50% |
| 1.4 | Linear SVM | 98.00% |
| 1.5 | Quadratic SVM | 97.70% |
| 1.6 | Cubic SVM | 97.00% |
| 1.7 | Fine Gaussian SVM | 80.70% |
| 1.8 | Medium Gaussian SVM | 97.90% |
| 1.9 | Coarse Gaussian SVM | 98.10% |
| 1.10 | Fine KNN | 79.80% |
| 1.11 | Medium KNN | 97.80% |
| 1.12 | Coarse KNN | 96.30% |
| 1.13 | Cosine KNN | 97.10% |
| 1.14 | Cubic KNN | 97.60% |
| 1.15 | Weighted KNN | 97.80% |
| 1.16 | Boosted Tree | 62.70% |
| 1.17 | Bagged Tree | 97.30% |
| 1.18 | Subspace Discriminate | 95.80% |
| 1.19 | Subspace KNN | 85.40% |
| 1.20 | RUSBoosted Tree | 62.70% |
| **1.21** | **Proposed IANN** | **98.80%** |

**Table 6** Comparison of breast cancer classification performance between existing methods in the literature and the proposed method.

| Author name and reference | Method | Accuracy |
|---|---|---|
| *Aalaei et al. (2016)* | GA- Based ANN | 96.50% |
| *Salama & Abdelhalim (2012)* | SMO | 97.70% |
| *Xue, Zhang & Browne (2014)* | PSO | 92.98% |
| *Nekkaa & Boughaci (2015)* | MA+SVM | 97.88% |
| *Quy et al. (2020)* | APSO-NN | 98.24% |
| *Amrane et al. (2018)* | KNN | 97.51% |
| *Abdar et al. (2020)* | SV-NB-3-MetaClassifier | 98.07% |
| **The proposed** | **IANN** | **98.80%** |

in both tabular and graphical form, proved that the proposed IANN greatly enhanced the overall classification performance by differentiating between normal and malignant tumors for superior breast cancer diagnosis. This study confirmed that the use of the Taguchi method with ANN improved classification performance relative to existing breast cancer classification methods. While the results are promising, future research can focus on feature selection to reduce the computational discrepancy. Also, future studies should

**Table 7  10-fold cross-validation experimental performance.**

| Cross validation (10-folds) | Training accuracy (%) | Test accuracy (%) |
|---|---|---|
| K-1 | 99.2 | 97.9 |
| K-2 | 98.6 | 100 |
| K-3 | 99.0 | 100 |
| K-4 | 98.6 | 98.2 |
| K-5 | 97.8 | 98.5 |
| K-6 | 98.2 | 98.2 |
| K-7 | 98.4 | 98.5 |
| K-8 | 98.0 | 98.6 |
| K-9 | 98.8 | 98.8 |
| K-10 | 98.3 | 98.2 |
| Average | 98.5 | 98.7 |

focus on the practical implications of Type I (FP) and Type II (FN) error in breast cancer classification. Meanwhile, an express medical test for breast cancer classification has often been designed in such a way that Type I error is given precedence. This is done to prevent a breast cancer patient from going about without the knowledge of having a malignant tumor.

### Funding

This research was supported by a grant from the Universiti Kebangsaan Malaysia (UKM), UKM Grant Code: GGP-2019-023). The funders had no role in study design, data collection and analysis, decision to publish, or preparation of the manuscript.

### Grant Disclosures

The following grant information was disclosed by the authors:
The Universiti Kebangsaan Malaysia (UKM): GGP-2019-023.

### Competing Interests

The authors declare there are no competing interests.

### Author Contributions

- Md Akizur Rahman conceived and designed the experiments, performed the experiments, analyzed the data, performed the computation work, prepared figures and/or tables, authored or reviewed drafts of the paper, and approved the final draft.
- Ravie chandren Muniyandi conceived and designed the experiments, performed the experiments, analyzed the data, prepared figures and/or tables, authored or reviewed drafts of the paper, arrange Funding, and approved the final draft.
- Dheeb Albashish performed the experiments, analyzed the data, prepared figures and/or tables, authored or reviewed drafts of the paper, and approved the final draft.

- Md Mokhlesur Rahman and Opeyemi Lateef Usman analyzed the data, prepared figures and/or tables, authored or reviewed drafts of the paper, and approved the final draft.

## Data Availability

Data is available at the Wisconsin Diagnostic Breast Cancer (WDBC): http://archive.ics.uci.edu/ml/datasets/Breast+Cancer+Wisconsin+%28Diagnostic%29.

Code is available as a Supplemental File.

## Supplemental Information

Supplemental information for this article can be found online at http://dx.doi.org/10.7717/peerj-cs.344#supplemental-information.

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
