# Peer review of "Artificial neural network with Taguchi method for robust classification model to improve classification accuracy of breast cancer"

_PeerJ Computer Science, doi:10.7717/peerj-cs.344_

## Round 0.1 · original submission · Minor Revisions

Both reviewers suggest work on results and presentation before second round.

Reviewer 1 ·

Basic reporting

The English in the report is generally of a high quality. However, there are a number of occasions where the authors confuse the technical terms with respect to the number of hidden layers, and the number of the neurons in a hidden layer, which causes significant confusion.

Literature is patchy. There are significant gaps in wider application of the technique of Taguchi to the design of Artificial Neural Networks (although there is considerable coverage of ANNs applied to Breast Cancer Screening). There are serious concerns regarding assertions made with respect to ‘killer diseases’ and breast cancer being the biggest cancer killer of women (WHO and CDC sources suggest that lung cancer is the biggest killer. The literature used by the author to support their claim does not appear to be peer-reviewed). One of the references that the cite as using ANNs to classify cancer actually describes biomarkers for identifying dyslexia.

Professional Structure could be improved. There is significant repetition throughout the document. There are too many figures, not all of which add to the narrative of the document, and some of the background information on Taguchi and ANNs would be better positioned as supplementary material (in my opinion).

The hypothesis is potentially self-contained and relevant but hasn’t been exploited to its full potential.

Results are clearly expressed. However, there are significant flaws in the experimental design which casts doubt on how meaningful they are.

Experimental design

The subject material is within scope for PeerJ Computer Science. The research question is relevant and sufficiently defined.

The investigation, however is not rigorous from a technical perspective. The application of Taguchi to identify the number of neurons does not appear necessary, as the authors consider only one parameter (the number of hidden neurons) and then only present results for half of the experiments suggested by their own Taguchi.

The methods are sufficiently detailed, although in my opinion are not applied appropriately.

Validity of the findings

It is my opinion that the findings of the analysis are not valid with respect to the original research intention.

Although the results show that the 15 neuron ANN outperforms other models with respect to correct classification of malignancies and non-malignancies, this is simply as a consequence of a brute force approach with respect to the number of hidden nodes and ignores other parameters in the ANN which may impact on optimisation, such as threshold function and momentum. It is not a result of a Taguchi analysis. The paper actually details the difference in performance of a 15 Neuron hidden layer ANN due to split between training, test and validation of the breast cancer dataset. It therefore does not demonstrate the power of Taguchi in the design of ANN, and so does not meet the objectives as stated in the introduction. Conclusions drawn which indicate otherwise are therefore spurious.

Additional comments

The issues that the authors set out to address are important and highlight the power of technology in supporting clinical decisions in complex, potentially life-threatening conditions. However, the methodological approach is flawed, and Taguchi hasn’t been applied appropriately, in my opinion. The paper shows that via an iterative approach, the authors have arrived at an ANN architecture that outperforms other validated models on an established dataset, and needs to be repositioned as such, It is misleading to the readers to state the design is a consequence of a robust Taguchi analysis.

·

Basic reporting

Overall Acceptable.

1) It would be better if the author could cite more recent published papers. Currently, there's no cited paper published after 2019.

2) The information density of table II is very low. The author can consider using figures instead of that table.

Experimental design

Acceptable

Validity of the findings

1) Line 337-338, "In addition, 70% of the dataset for training, 15% for validation, and 15% for testing is the optimal partitioning…". The choice of the data partition benefits the model proposed by the author. It is unclear that the results are valid due to the selection bias risk. It would be better if the author could compare the proposed model to other existing classification models with other data partitions.

2) Table III shows the results from 30 different classification simulations using the WDBC dataset with their corresponding percentage errors. The author presents the average of the tests' accuracy. However, the variance of the results is also worth showing. This is also a common problem for other tables.

3) Line 257-258, "A two-layer feed-forward neural network, Gradient Descent with Adaptive Learning Rate (GDA) optimizer with Sigmoid activation function and Softmax Output Neurons, was used to classify the datasets with adequate neurons in the hidden layer." The author uses the GDA optimizer with Sigmoid activation function and Softmax Output Neurons as the model's components. The reason for choosing them is not mentioned in the paper. Adaptive gradient descent algorithms such as Adagrad, Adadelta, RMSprop, and Adam are state of the art and more acceptable optimizers. Also, there could be different activation and output function. It would be better to discuss the choice.

---

## Round 0.2 · accepted · Accept

The revision process is now finished and independent Reviewers recommend your paper for further processing, therefore I am pleased to forward your paper to the editorial office with a positive recommendation.

·

Basic reporting

Acceptable

Experimental design

Acceptable

Validity of the findings

Acceptable